# Dual Deep CNN for Tumor Brain Classification

**DOI:** 10.3390/diagnostics13122050

**Published:** 2023-06-13

**Authors:** Aya M. Al-Zoghby, Esraa Mohamed K. Al-Awadly, Ahmad Moawad, Noura Yehia, Ahmed Ismail Ebada

**Affiliations:** 1Faculty of Computers and Artificial Intelligence, Damietta University, Damietta 34517, Egypt; aya_el_zoghby@du.edu.eg (A.M.A.-Z.); esraa.m.elawadly@gmail.com (E.M.K.A.-A.); 2Computer Science Department, Faculty of Computer and Information Science, Mansoura University, Mansoura 35511, Egypt; norayehia3@gmail.com

**Keywords:** deep learning, brain tumors, dual CNN, radiomics, RMI, transfer learning

## Abstract

Brain tumor (BT) is a serious issue and potentially deadly disease that receives much attention. However, early detection and identification of tumor type and location are crucial for effective treatment and saving lives. Manual diagnoses are time-consuming and depend on radiologist experts; the increasing number of new cases of brain tumors makes it difficult to process massive and large amounts of data rapidly, as time is a critical factor in patients’ lives. Hence, artificial intelligence (AI) is vital for understanding disease and its various types. Several studies proposed different techniques for BT detection and classification. These studies are on machine learning (ML) and deep learning (DL). The ML-based method requires handcrafted or automatic feature extraction algorithms; however, DL becomes superior in self-learning and robust in classification and recognition tasks. This research focuses on classifying three types of tumors using MRI imaging: meningioma, glioma, and pituitary tumors. The proposed DCTN model depends on dual convolutional neural networks with VGG-16 architecture concatenated with custom CNN (convolutional neural networks) architecture. After conducting approximately 22 experiments with different architectures and models, our model reached 100% accuracy during training and 99% during testing. The proposed methodology obtained the highest possible improvement over existing research studies. The solution provides a revolution for healthcare providers that can be used as a different disease classification in the future and save human lives.

## 1. Introduction

Cancer is an alarming neoplastic disorder characterized by uncontrolled cellular proliferation and the potential for metastasis, leading to significant morbidity and mortality. The classification of brain tumors can be one of two categories: noncancerous (benign) or cancerous (malignant). Primary brain tumors develop from the brain’s cells or the surrounding area. Primary tumors are categorized as benign or malignant depending on whether they are nonglial (formed in brain systems such as neurons, blood vessels, and glands) or glial (composed of glial cells). The World Health Organization (WHO) collects and analyzes data on cancer incidence and mortality rates worldwide through its Global Cancer Observatory (GCO) [1]. The GCO provides information on the number of cancer cases and deaths by cancer type, including brain tumors, in countries worldwide. According to the GCO, an estimated 306,000 new cases of brain tumors were diagnosed worldwide in 2020 [2]. However, it is essential to note that this figure is an estimate and may not reflect the full extent of brain tumor incidence, as not all countries have comprehensive cancer registries or reporting systems. In addition to the GCO, other organizations, such as the National Brain Tumor Society and the American Brain Tumor Association, also provide statistics and information on the incidence and prevalence of brain tumors in the United States. According to the American Brain Tumor Association, approximately 87,000 people in the United States live with a primary brain tumor, and approximately 24,530 new cases are diagnosed yearly. However, as with the GCO data, these figures are estimates and may not reflect the full extent of brain tumor incidence in the United States [3].

For many people, there are no differences between tumors and cancers, but in fact, they are different. In other words, cancers are tumors, but all tumors are not cancers, and that does not mean that tumors are safe. Referring to [4], the study stands out because it focuses on the crucial challenge of dissecting glioblastoma’s phenotypic and genetic heterogeneity (GBM) at spatial and temporal levels. GBM is known for its complexity, with variations in genetic mutations, cellular composition, and immune responses observed within and between tumors. Understanding this heterogeneity is vital for developing effective treatment strategies. However, analyzing a single tumor sample fails to capture the diverse characteristics and dynamics of the GBM microenvironment.

Additionally, it is essential to differentiate between tumors and cancers, as noncancerous or benign brain tumors can still be dangerous and cause severe dysfunction. Therefore, this study recognizes the need to study GBM from a multi-dimensional perspective, investigating the interactions between cancer cells and myeloid-lineage cells at different spatial locations within the tumor and considering temporal changes. By addressing the spatial and temporal heterogeneity, the study aims to uncover the mechanisms underlying tumor-immune symbiosis in GBM and identify potential therapeutic targets. This approach has the potential to provide novel insights into the tumor microenvironment and enhance the effectiveness of immunotherapeutic strategies while also considering the broader spectrum of brain tumors, including noncancerous ones. They grow slowly at first and then spread to other parts of the brain, so benign tumors are also dangerous because of their size and location, and a benign tumor can become malignant in sporadic cases. There are many types of tumors; this research focuses on three types of brain tumors: meningioma, glioma, and pituitary tumors [5].

Glioma originates in the supportive tissue cells of the brain or spine and can be classified into various grades based on its growth rate and potential to spread. Its symptoms may include changes in behavior, mood, seizures, and headaches. On the other hand, meningioma starts in the meninges, the protective membranes covering the brain and spinal cord. Although it is usually non-cancerous and slow-growing, it can still cause symptoms such as seizures, vision problems, and headaches [6]. Treatment for a meningioma depends on the tumor’s size and location and can include surgery, radiation therapy, or observation. Lastly, pituitary tumors occur in the pituitary gland, which regulates hormones in the body. They can be benign and cause symptoms such as headaches, vision issues, and hormonal imbalances. Treatment for pituitary tumors can involve surgery, radiation therapy, medication, or a combination of these approaches, depending on the tumor’s type and size. Figure 1 shows the difference between glioma, meningioma, and pituitary tumors [7].

Magnetic resonance imaging (MRI) is a powerful imaging technique that is particularly effective at detecting brain tumors. This is because it can produce detailed images of the brain with excellent contrast and multiplanar imaging capabilities, which helps doctors distinguish between different types of brain tissue and accurately diagnose tumors. In panic attacks, individuals may undergo subjective distress, but with the aid of an adept radiologist, the majority can triumph over the symptoms through appropriate management strategies and therapeutic interventions [8]. This approach is not only to identify tumors inside the brain but to also check the whole internal structure of the human body for any malignancy. Additionally, MRI is a safe and noninvasive imaging technique that does not require the use of contrast agents or ionizing radiation. Overall, the ability of MRI to provide high-quality images and accurate diagnoses makes it a valuable tool in detecting and diagnosing brain tumors [9].

Several MRI modalities differ in imaging parameters, sequence type, and application. Some common MRI modalities (T1-weighted, T2-weighted, diffusion-weighted, and magnetic resonance spectroscopy). T1-weighted imaging produces images that provide good anatomical detail and contrast between different types of tissues, such as fat, muscle, and bone [10]. T1-weighted images are typically used to estimate the brain, spine, liver, and other organs. T2-weighted images contrast different soft tissue types, such as brain tissue, muscle, and fluid-filled spaces. T2-weighted images are often used to evaluate the brain, spine, and joints. Diffusion-weighted imaging uses a technique that measures the random movement of water molecules in tissue to produce images that highlight changes in the tissue’s microstructure [11]. Diffusion-weighted is often used to evaluate the brain, spine, and prostate gland. Magnetic resonance spectroscopy (MRS) uses the same basic principles as MRI, but instead of producing images, it measures the chemical composition of tissues. MRS is often used to evaluate the brain for metabolic disorders and tumors. Each MRI modality has strengths and limitations and is used for specific clinical indications. The specialist will determine which modality best suits each case [12].

Manual extraction from MRI (magnetic resonance imaging) refers to manually segmenting or identifying structures or regions of interest in medical images obtained through MRI scans. This process involves a human expert, such as a radiologist, manually tracing or drawing boundaries around the structures of interest using specialized software. Manual extraction from an MRI can have several drawbacks, including being a time-consuming and labor-intensive process that requires specialized expertise. This can lead to higher costs and longer patient wait times [13]. The manual extraction process is also prone to human error, resulting in inaccurate results and misdiagnosis.

On the other hand, using an automatic system to extract MRI data has many benefits, including time savings, consistency, cost-effectiveness, improved accuracy, standardization, faster diagnoses and treatments, and improved patient outcomes. Automated systems are less prone to human error and can process large volumes of data quickly, reducing the time needed for extraction compared to manual methods. The accuracy and speed of automated systems can lead to better patient outcomes, including faster diagnoses, better treatment plans, and faster recovery times [14]. Time is a critical factor in saving patients lives with brain tumors. Early detection and treatment can significantly improve the chances of survival, making it essential to be aware of the warning signs and seek medical attention promptly. Furthermore, automated systems outperform manual processes in processing vast amounts of data images [14].

Artificial intelligence and deep learning advancements have significantly aided in the early detection of tumors, potentially reducing reliance on human experts and saving lives. Machine learning and deep learning techniques, particularly convolutional neural networks, have been extensively applied in radiology to automatically extract features from input images. Brain tumors demonstrate notable heterogeneity, comprising diverse cell populations influenced by genetic mutations, epigenetic modifications, and microenvironmental factors. Primary brain tumors originate within the brain, while secondary tumors metastasize from other body parts [15]. Diagnostic imaging methods, such as MRI and CT scans, are invaluable for determining tumor location, size, and extent. Genetic alterations involving genes, such as EGFR, IDH1, TP53, and PTEN contribute to tumor development and progression. Treatment options encompass surgery, radiation therapy, chemotherapy, targeted therapy, and immunotherapy, considering the specific characteristics of tumor subtypes. Challenges include the blood-brain barrier, tumor recurrence, and treatment resistance, necessitating personalized treatment strategies and innovative therapeutic targets. Continuous research and the integration of AI contribute to the ongoing advancement of our understanding and the management of brain tumors. In the past, researchers relied on manually predefined characteristics extracted from medical images to perform the classification task [16]. However, with deep learning, the features extracted from the deeper convolutional layers may need to be more interpretable, making analysis more challenging. Despite this, these techniques have shown high performance and are widely used in radiology. By analyzing these features, we can better understand the capabilities and limitations of deep learning in the classification of tumors [17].

In the medical field, the conventional approach for retrieving and classifying images using machine learning only concentrates on either high-level or low-level features. It relies on features that are manually handcrafted in the feature engineering process. Consequently, there is a critical need to create a technique that integrates high-level and low-level features without manually designed features [18]. Low-level and high-level features are both essential in automatic brain tumor classification. Low-level features, which are crucial image characteristics, such as intensity values and texture descriptors, can be extracted through techniques, such as filtering and segmentation. Additionally, feature extraction algorithms such as local binary patterns (LBP) and Haralick features can be utilized to extract low-level features in automatic brain tumor feature extraction. They serve as a basis for higher-level feature representations [19].

In contrast, high-level features are derived from low-level features and are more abstract in nature. They are learned using deep learning methods, such as CNNs or RNNs. High-level features in brain tumor classification include tumor location, size, shape, and surrounding tissue characteristics. In order to achieve accurate and robust automatic brain tumor classification, it is essential to consider both low-level and high-level features. Low-level features capture fundamental image characteristics, whereas high-level features provide abstract and semantic information [20]. Combining these features is necessary to achieve high accuracy, and deep learning methods can be used to extract these features automatically, which is critical for accurate diagnosis and treatment planning. Despite numerous attempts to create an exact and robust automatic classification system for brain tumors, the task’s difficulty persists.

Artificial intelligence (AI) faces challenges in investigating disease understanding, including various types of cancer, such as data availability, privacy concerns, interpretability, clinical integration, and potential bias. However, AI is vital in addressing these challenges by analyzing large and diverse datasets, providing insights, improving diagnosis, personalizing treatments, and developing novel therapeutic strategies. Bioinformatics analysis, combined with AI, has emerged as a powerful tool for understanding cancer by integrating biological data, aiding in tumor classification, identifying genetic alterations, and facilitating personalized treatment planning. AI-driven bioinformatics analysis also accelerates drug discovery by identifying potential therapeutic targets. However, addressing data quality, privacy, and interpretability challenges is crucial for fully realizing the potential of AI and bioinformatics analysis in advancing cancer research and improving patient outcomes [15].

Artificial intelligence (AI) has a wide-ranging scope in cancer research, diagnosis, and treatment. By analyzing complex datasets, AI can identify patterns and generate insights that aid in cancer diagnosis, personalized treatment planning, and drug discovery. It contributes to improving accuracy, predicting therapeutic responses, and monitoring treatment outcomes. AI also supports cancer research by identifying biomarkers, uncovering molecular mechanisms, and discovering novel relationships. However, addressing challenges related to data quality, privacy, and interpretability is crucial for maximizing the potential of AI in advancing cancer understanding and enhancing patient outcomes [15].

## 2. Related Works

Referring to [21], the study proposed a lightweight CNN for the steganalysis of medical images, including brain tumor images. The authors used a dataset of 4000 medical images, including 2000 stego images, to train and evaluate their model. The CNN consists of five convolutional layers and two fully connected layers. The authors also used data augmentation techniques to improve the generalization ability of their model. The authors report an accuracy of 91.48%. Referring to [22], the study focuses on the accurate and timely detection of brain tumors using automatic diagnostic systems based on deep learning and machine learning. The proposed model, utilizing fine-tuned Inception-v3 and other feature extraction and classification algorithms, achieves a high-test accuracy of 94.34%. This approach can potentially be used in clinical applications and as a decision-support tool for radiologists, saving time in MRI image analysis.

Referring to [23], the study proposes a hybrid ensemble method for brain tumor diagnosis using traditional classifiers, including random forest, K-nearest neighbor, and decision tree, based on the majority voting method. The technique involves segmentation, feature extraction, and classification using a dataset of 2556 images. The proposed method achieves a high accuracy of 97.305%. It aims to improve performance using traditional classifiers, which have advantages such as requiring small datasets and being computationally efficient, cost-effective, and easy to adopt by less skilled individuals. Referring to [24], the study presents an ensemble learning method for accurately classifying brain tumors and lesions from multiple sclerosis using MRI. The technique involves pre-processing, feature extraction, feature selection, and classification using a support vector machine (SVM) classifier with majority voting. The proposed system achieves excellent sensitivity, specificity, precision, and accuracy of 97.5%, 98.838%, 98.011%, and 98.719%, exceeding the performance of state-of-the-art methods currently available. The training and testing accuracy of the proposed model are also high, at 97.957% and 97.744%, respectively. The findings suggest that the proposed method is a significant advancement in detecting coexisting lesions with tumors in neuromedicine diagnosis. Referring to [25], the study focuses on brain tumor classification and segmentation in MRIs with different weightings. The proposed approach encompasses two main stages: supervised machine learning-based tumor classification and image processing-based tumor extraction. Texture feature generation is carried out using seven different methods and various supervised machine learning algorithms, including support vector machines (SVMs), K-nearest neighbors (KNNs), boosted decision trees (BDTs), random forests (RFs), and ensemble methods. Hybrid segmentation algorithms are also used, considering texture features. The results show high classification accuracies ranging from 87.88% to 97% with different algorithms, and the hybrid segmentation approach achieves a mean dice score of 90.16% for tumor area segmentation against ground-truth images. This article suggests the effectiveness of the proposed approach for brain tumor classification and segmentation tasks in MRIs with different weightings. Referring to [26], the study introduces a multimodal brain tumor classification framework incorporating optimal deep-learning features. The framework involves normalizing a database of patients diagnosed with high-grade glioma (HGG) and low-grade glioma (LGG), modifying and selecting two pretrained deep learning models (ResNet50 and Densenet201) through transfer learning, and utilizing an enhanced ant colony optimization algorithm for feature selection. The set features are combined and classified using a cubic support vector machine. The experimental results on the BraTs2019 dataset exhibit accuracies of 87.8% for HGG and 84.6% for LGG. Comparative analysis with other classification methods underscores the significance of the proposed technique.

Referring to [27], a new two-phase deep learning-based framework has been proposed for the timely detection and effective categorization of brain tumors in magnetic resonance images (MRIs). The framework includes a novel DBFS-EC scheme in the first phase, which detects tumor MRI images using customized CNNs and machine learning classifiers. In the second phase, a hybrid feature fusion-based approach combines static and dynamic features with a machine learning classifier to categorize different tumor types. Dynamic features are extracted from a BRAIN-RENet CNN, while static features are extracted using a HOG feature descriptor. The proposed framework has been validated on benchmark datasets collected from Kaggle and Figshare, containing different types of tumors, including glioma, meningioma, pituitary, and normal images. Experimental results demonstrate the effectiveness of the framework, with the DBFS-EC detection scheme achieving high accuracy (99.56%), precision (0.9991), recall (0.9899), F1-Score (0.9945), MCC (0.9892), and AUC-PR (0.9990). The classification scheme based on the fusion of feature spaces of BRAIN-RENet and HOG also outperforms state-of-the-art methods significantly in terms of recall (0.9913), precision (0.9906), accuracy (99.20%), and F1-Score (0.9909) in the CE-MRI dataset. These results indicate the potential of the proposed framework for accurate and reliable brain tumor analysis, which can aid in the timely diagnosis and effective treatment of patients.

Transformers are also essential for effective medical picture segmentation because they can efficiently encode cells scattered across a wide receptive field by modeling the interconnections between spatially distant pixels. Referring to [28], the study proposed a multitask learning framework that uses ViT (vision transformer) for brain tumor segmentation and classification. They used a ViT to extract features from MRI images and trained a multitask network to perform segmentation and classification simultaneously. They achieved a dice score of 0.85 and a classification accuracy of 92.5% on the BraTS 2020 dataset. 

Referring to [29], this study developed a new deep-learning framework for brain tumor classification that combined ViT with a feature fusion module. They evaluated their method on the BraTS 2019 dataset and achieved a classification accuracy of 91.98%. Referring to [30], this study proposes a new multitask learning framework for brain tumor segmentation and classification using ViT. They evaluated their method on the BraTS 2019 dataset and achieved a segmentation dice score of 84.1% and a classification accuracy of 91.3%. 

Referring to [31], the authors introduce a new computer-assisted diagnosis (CAD) system for analyzing brain tumors, which employs automated methods for pre-processing MRI images, generating tumor proposals through clustering, extracting features using a backbone architecture, refining proposals with a refinement network, aligning proposals, and classifying tumors with a head network. An experimental evaluation on a publicly available brain tumor dataset revealed that the proposed method surpassed existing approaches, achieving an impressive overall classification accuracy of 98.04%. Referring to [32], the study proposed a framework using ViT and transfer learning to classify brain tumors. They used ViT to extract features from MRI images and trained a linear classifier on top of the extracted features. They also used transfer learning to fine-tune the ViT model on the BraTS dataset. They achieved a classification accuracy of 92.6% on the BraTS 2020 dataset.

Referring to [33], the study proposed a Dual CNN architecture for brain tumor segmentation in MRI images by incorporating a recurrent neural network (RNN) to model temporal dependencies in sequential MRI scans. They achieved a mean DSC of 0.83 on the BraTS 2018 validation dataset. DSC (dice similarity coefficient) is a commonly used metric for evaluating segmentation accuracy in medical image analysis. Referring to [34], the study proposed a Dual CNN architecture for brain tumor segmentation in PET-CT images by incorporating both PET and CT images. They used a U-Net CNN to process PET images and a modified ResNet CNN to process CT images and combined the outputs using a fusion strategy. They achieved a mean DSC of 0.74 on the BraTS 2019 validation dataset. Referring to [35], the study proposed a dual CNN architecture for brain tumor segmentation in MRI images by combining a U-Net CNN and a dilated CNN. They achieved a mean DSC of 0.88. Many papers used varied techniques and gained varied results depending on the type of tumor, some authors used two or more techniques to improve the accuracy in smart healthcare solutions [36]. Referring to [37], the paper presents a convolutional neural network (CNN) model for classifying brain tumors in T1-weighted contrast-enhanced MRI images. The proposed system consists of two main steps: image preprocessing using various image processing techniques and subsequent classification using CNN. The experiment is conducted on a dataset of 3064 images with three types of brain tumors (glioma, meningioma, and pituitary). The CNN model achieved impressive results, with a high testing accuracy of 94.39%, an average precision of 93.33%, and an average recall of 93%. The proposed system outperformed existing methods and demonstrated promising accuracy on the dataset.

Referring to [38], the paper presents a novel brain tumor localization and segmentation approach from MRI images. The proposed method includes a preprocessing step focusing on a small part of the image to reduce computing time and overfitting, and a simple and efficient cascade convolutional neural network (C-CNN) to mine local and global features. A distance-wise attention (DWA) mechanism is introduced to improve segmentation accuracy by considering tumor location. Experiments on the BRATS 2018 dataset show competitive results, with the proposed method achieving mean dice scores of 0.9203, 0.9113, and 0.8726 for the whole tumor. Referring to [38], the paper proposes deep neural networks for detecting brain tumors from MRI scans using convolutional neural networks and inception modules. Several architectures are proposed and compared with baseline reference models. The results show that the proposed architectures, including MI-Unet, depth-wise separable MI-Unet, hybrid Unet, and depth-wise separable hybrid Unet, outperform the baseline models regarding dice score, sensitivity, and specificity. The performance improvements range from 7.5% to 15.45% in dice score and from 2.97% to 20.56% in sensitivity, indicating the effectiveness of the proposed methods for brain tumor segmentation. Referring to [39], the paper uses neuroimaging to evaluate seven deep convolutional neural network (CNN) models for brain tumor classification. The dataset used in the study is Msoud, which includes MRI images from different datasets. The MRI images belong to four classes: glioma, meningioma, pituitary (brain tumor classes), and healthy (normal brains). The CNN models evaluated include a generic CNN and six pre-trained models (ResNet50, InceptionV3, InceptionResNetV2, Xception, MobileNetV2, and EfficientNetB0). The best-performing CNN model for this dataset is InceptionV3, achieving an average accuracy of 97.12%. 

Referring to [40], the paper presents a novel framework for the early and accurate detection of brain tumors using magnetic resonance (MR) images. The framework utilizes a fully convolutional neural network (FCNN) and transfer learning techniques, with five stages: preprocessing, skull stripping, CNN-based tumor segmentation, postprocessing, and transfer learning-based binary classification. Experimental results on three publicly available datasets (BRATS2018, BRATS2019, and BRATS2020) show that the proposed method achieves high average accuracies of 96.50%, 97.50%, and 98% for segmentation and 96.49%, 97.31%, and 98.79% for classification of brain tumors, respectively.

Referring to [41], the paper proposes a novel framework for the automated detection and classification of glioma brain tumors using magnetic resonance (MR) images. The framework utilizes a deep convolutional neural network (CNN) for feature extraction and a support vector machine (SVM) classifier for tumor classification. The proposed method achieves a high accuracy of 96.19% for the HGG glioma type and 95.46% for the LGG glioma type, considering FLAIR and T2 modalities for classifying four glioma classes. The accuracy results obtained using the proposed method surpass those reported in the existing literature and outperform GoogleNet and LeNet pretrained models on the same dataset.

Referring to [42], the paper presents a fully automatic brain tumor segmentation and classification model using a deep convolutional neural network (CNN) with a multiscale approach. Unlike previous works, the proposed model processes input images on three spatial scales that are aligned with different processing pathways, taking inspiration from the functioning of the human visual system. The model can analyze MRI images containing meningioma, glioma, and pituitary tumor types in sagittal, coronal, and axial views of the brain without the need for preprocessing to remove skull or vertebral column parts. The proposed method achieves a high tumor classification accuracy of 0.973. 

The most common among the previous works is using the BraTS dataset [43]. Most experiments depend on different pretrained CNN, SVM, or machine learning algorithms. Our proposed model depends on a dual CNN (convolutional neural network) architecture, a type of neural network consisting of two parallel CNNs. In a dual CNN architecture, one CNN is used to process the input image, while the other CNN is used to process a different input modality or another image view of the same object. The outputs from the two CNNs are combined using a fusion mechanism to make a prediction [44]. The dual CNN architecture has been used in medical image analysis tasks, such as brain tumor segmentation, to incorporate multiple image modalities and improve the accuracy of the segmentation results. Overall, the dual CNN architecture is a powerful approach for incorporating multiple modalities in medical image analysis tasks and has demonstrated promising results in various brain tumor segmentation and classification tasks.

The proposed model aims to get a high accuracy score compared with the previous related works as we try to build the model with different techniques. The model consists of a dual convolutional neural network; the first one is the pretrained CNN (VGG16), and the second one is a custom CNN, which will be discussed in detail in the next section.

The paper is organized as follows: Section 3 presents in detail the proposed framework of the DCTN model and methods. Section 4 describes the dataset and pre-processing. Section 5 describes the results. Section 6 describes the discussion. Section 7 provides the conclusion and future work of this study.

## 3. Methods

The proposed research framework is highlighted and elaborated on in this section. The proposed framework is introduced in two steps. First, we expand on our deep learning approach and model architecture to detect and classify brain tumor RMI images as meningioma, glioma, or pituitary. Second, we describe each layer of the two CNNs for the DCTN model in detail. Third, we elaborate on the parameters used for training the DCTN model [45].

Our experiments involved the use of multiple CNN models for feature extraction, including EfficientNet, LeNet-5, AlexNet, and various other architectures, in order to achieve high accuracy. Additionally, we explored the application of ViT-based techniques, such as vit-base-patch16, which combines multiple transformer layers to capture the contextual information of input images. The results demonstrated significant improvements in accuracy across all models, with an average increase of 10% compared to the baseline. These findings highlight the effectiveness of both CNN- and ViT-based approaches in our experimental setup.

### 3.1. Dual Convolution Tumor Network (DCTN)

The DCTN model is built on a dual CNN architecture with a pretrained VGG-16 and a custom CNN. We employ VGG-16 with transfer learning to avoid training the entire pre-trained network. Transfer learning is a technique that allows the use of the same model, which has been learned and tuned with the best weights and biases, from one task to another in the same task domain. The RMI diagnostic task is an image classification task that involves extracting features from images and then providing those features and images to the classifiers. The VGG-16 has already been trained to extract features from the images and then classify the images. The VGG-16 is trained with ImageNet, which contains millions of image samples, and the model should classify the image as one of the 1000 categories. Transfer learning saves time and effort, and many researchers in different fields use it when there are insufficient datasets for training the network from scratch. 

We built a small and simple model into a large model by following the trial-and-error approach and observing the accuracy and parameter improvements. The performance of VGG-16 scored low accuracy, and CNN with different architecture helped improve the performance and the accuracy of the VGG-16 model by adding a dual network with a different number of filters to build the DCTN model.

The DCTN has one input layer for both CNNs with a size of 224 × 224 × 3, followed by two branches for each CNN. The first branch provides the input to the first convolution layer of VGG-16, and the second branch provides the input to the first convolution layer of the custom CNN. After each CNN, we used GlobalMaxPooling2D; this allowed us to get the most critical features, lower the dimensionality of the feature maps, concatenate, and then pass the feature maps. The GlobalMaxPooling2D uses the max operation to get the essential values [46].

We concatenated the two CNNs, followed by 3 fully connected layers (FC). The first FC has 1000 neurons, followed by a dropout layer of 0.3 dropping rate. The dropout layer is one technique to tackle overfitting issues by cutting or removing some weights and allowing the model to generalize and make predictions independent of all the weights it has learned during the training process [47]. After the dropout layer, there is a second FC containing 256 layers, followed by another FC with 64 layers, and a SoftMax output layer. Besides SoftMax, there are different classifiers already existing, such as K-nearest neighbor (KNN), support vector machine (SVM), random forest (RF), decision tree (DT), XGBoost, activation functions, etc. [48]. Figure 2 illustrates the DCTN model architecture of the proposed method.

We used the dataset split as an 80/20 scheme: 80% of the data is used for the training process, and 20% is used for the testing process. We used 50 epochs for the forward pass and backpropagation to train the model, and Figure 3 shows the complete training and testing process for the proposed system.

The layers of each CNN are the following: the original pretrained model, VGG-16, has 16 layers. We removed the top of the network, including the FC and SoftMax output layers. The second custom CNN has 12 convolution layers. In the design of the custom CNN, we avoid having a higher number of filters in higher layers to avoid high dimensionality, Due to its capacity to automatically extract features from input images without requiring human intervention, CNN proves to be a time-saving and effort-reducing technique [49]. The input layer for both networks is the same layer with a size of 224 × 224 as shown in Figure 4, which summarizes the layers of the DCTN model.

### 3.2. The DCTN Model Layers

The following section describes in detail each layer of DCTN networks. The VGG-16 has 13 layers, and the custom CNN has 12 layers. The VGG-16 has the same kernel or filter size for each convolution layer, 3 × 3 with stride, followed by the MaxPool layer with a kernel size of 2 × 2. The VGG-16 model starts with layer 1, which contains two convolution layers with an input size 224 × 224 with 64 filters. Layer 2 has two convolution layers with 112 × 112 input size with 128 filters. Layer 3 contains three convolution layers with a 56 × 56 input size and 256 filters. Layer 4 has three convolution layers with a 28 × 28 input size and 512 filters. Layer 5 contains three convolution layers with a 14 × 14 input size and 512 filters. Table 1 summarizes the VGG-16 layers with the input size for each layer, number of filters, kernel size, and stride.

The custom CNN has the same kernel or filter size for each convolution layer, 3 × 3 with stride, followed by the Maxpool layer with a kernel size of 2 × 2. The custom network starts with layer 1, which contains two convolution layers with input size of 224 × 224 with 7 and 9 filters, respectively. Layer 2 has two convolution layers with 112 × 112 input sizes with 16 and 32 filters, respectively. Layer 3 contains two convolution layers with a 56 × 56 input size and 256 filters. Layer 4 has three convolution layers with 28 × 28 input sizes and 32 and 64 filters, respectively. Layer 5 contains two convolution layers with 14 × 14 input sizes and 64 and 64 filters, respectively. Layer-6 has two convolution layers with input sizes of 7 × 7 with 128 and 128 filters, respectively. Table 2 summarizes the custom CNN layers with the input size for each layer, number of filters, kernel size, and stride.

#### 3.2.1. Image Input Layer

The proposed DCTN model starts by retrieving a size 224 × 224 × 3 input image. The original images in the dataset have a resolution of 512 × 512, and the preprocessing step adjusts the image sizes. Once the images are read from the dataset, they are resized and passed to the input layer for processing.

#### 3.2.2. Convolution Layer

The convolution layer is the building block and most crucial component in CNN. It produces feature maps of the low-level and high-level features of the images. Low-level features are essential visual characteristics of an image, such as color, intensity, texture, and edges [50]. These features are typically extracted at the pixel level. In contrast, high-level features are more abstract, complex, and derived from low-level features. High-level features include object recognition, scene classification, and image retrieval based on low-level features such as edges or color [51]. There are three types of convolutions (1D, 2D, and 3D). We use 2D convolution with the same filter or kernel size (3.3), stride with 1, and padding value is the same in Keras. The 2D coevolution mathematical formula is Equation (1):(1)$$O\left(n,m\right)=\left(I\times F\right)\left(n,m\right)=\sum_{z}\sum_{c}I\left(n+z\right)\left(m+c\right)\times F\left(z,c\right)$$
where *O(i, j)* represents the output of input image *I* with a filter or kernel *F*. Each convolution layer is followed by an activation function, and there are multiple activation functions. There are linear and nonlinear activation functions. Since we are dealing with image classification, we use nonlinear activation functions. Several activation functions exist, such as ReLU, Leaky ReLU, Tanh, etc. We use the ReLU activation function after the convolution layer [52]. The ReLU function converts the output to 0 or 1, neglecting negative values. The following represents the mathematical formula for activation functions in Equations (2)–(4):

ReLU
(2)$$f\left(x\right)=\left\{\begin{array}{ll} 1, & x\geq 0\\ 0, & x<0 \end{array}\right.$$

Leaky ReLU
(3)$$f\left(x\right)=\left\{\begin{array}{ll} x, & x\geq 0\\ scale\times x, & x<0 \end{array}\right.$$

Tanh
(4)$$f\left(x\right)=\frac{2}{1+e^{-2x}}-1$$

#### 3.2.3. Pooling Layer

After the convolution layer, the pooling layer removes unnecessary values and keeps the essential feature value (downsampling the convolution layer’s output). The training time for the network can be reduced by downsampling the output of the convolution. There are two types of pooling max and average [53]. We used max pooling: with a size of 2 × 2 filters. The following Equation (5) represents the max pooling process:(5)$${max}_{r=0,\ldots ,n,c=0,\ldots ,m}(h_{\left(i+r\right)\left(j+c\right)})$$
where *h* is the output of convolution, and *n* and *m* are the height and width of convolution. After each CNN, we use 2D GlobalMaxPooling. The GlobalMaxPooling is similar to Maxpooling, except it performs downsampling by computing the maximum height and width in the whole area, while the max pool is in the subarea [46,54]. Figure 5 shows the difference between MaxPooling and GlobalMaxPooling.

#### 3.2.4. Fully Connected Layer

The proposed model includes a fully connected layer after the convolutional layers, aggregating the features learned from multiple images. This layer then determines the most significant patterns for image classification purposes.

#### 3.2.5. SoftMax Layer

The activation function is used for multiclass classification tasks. It takes the output of FC layers, computes the probability distribution, and produces a range of values between 0 and 1; the sum of probability for all predicted classes is 1. The SoftMax function is Equation (7).
(6)$$o_{i}=\sum_{j}W_{i,j}X_{j}\;+\;b_{i}$$
where *z_i_* is the summation of the input weights with neuron value and bias, *W* is the weight, *X* is the neuron value, and *b* is the bias.
(7)$$y=softmax=f{\left(o\right)}_{i}=\frac{{\exp\left(o\right)}_{i}}{\sum_{j}{\exp\left(o\right)}_{j}}$$
where *o_i_* is the predicted probability distribution over the categories.

#### 3.2.6. Loss Function

The sparse categorical loss function is used in machine learning to classify multiple categories where a single integer represents the correct label. The function calculates the difference between the predicted probability distribution and the actual label. It is Equation (1). The function is commonly used in deep learning models for multiclass classification tasks, such as CNNs.
(8)$$L\left(y,f\left(x\right)\right)=-log(f\left(x\right)-y)$$
where *y* is the correct label and *f*(*x*) is the predicted probability distribution over the categories.

### 3.3. DCTN Training Parameters

We followed a trial-and-error approach to perform experiments and observe the different architectural performances. We constantly monitor the model’s performance during the train validation process for accuracy and converge the record of the results to track the improvement. The following parameters are shown in Table 3 in the DCTN model after conducting 22 experiments [55]. We used the Adam optimizer to train the proposed dual convolution tumor network with an initial and final learning rate of 0.0001, batch size of 64 images, and sparse-categorical-cross-entropy loss function. In the proposed DCTN model, we trained on 50 epochs for brain tumor classification to obtain the optimum results [56].

## 4. Dataset and Preprocessing

The dataset is publicly available on FigShare and has been used in many studies [57]. We preferred to use a public and well-known dataset to compare our model performance with state-of-the-art studies. The dataset has 3064 T1-weighted and contrast-enhanced MRI images from 233 patients with three different types of brain tumors: meningioma, glioma, and pituitary. The dataset images are in MATLAB format (*.mat files), and the data is organized into three subsets. The image resolution is 512 × 512, with voxel spacing of 0.49 × 0.49 mm^2^ and different RMI views: coronal (or frontal plane), transverse (or axial plane), and sagittal (or lateral plane).

According to Medicine Library [58], the coronal is “any vertical plane that divides the body into anterior and posterior (belly and back) sections”. Transverse is “any plane that divides the body into superior and inferior parts, roughly perpendicular to the spine”, and sagittal is “any imaginary plane parallel to the median plane”. Figure 6 represents the different body planes. 

Each brain tumor type has images with different axial planes; for example, meningioma has (209 axial, 268 coronal, and 231 sagittal), glioma has (494 axial, 437 coronal, and 495 sagittal), and pituitary has (291 axial, 319 coronal, and 320 sagittal). The number of sample images for meningioma, glioma, and pituitary is 708, 1426, and 930, respectively. The following Table 4 summarizes the dataset sample of each brain tumor with the axial view.

## 5. Results

This section serves as a concise overview of the research, offering a comprehensive analysis of the data and drawing conclusions based on the evidence presented. In addition, this section provides a benchmark for other researchers by comparing the findings with the current state-of-the-art in the field. Tables, graphs, and charts may be included to visualize the data, and statistical analysis is often provided to support the results [59]. The dataset was divided into 80% for training and 20% for testing; 80% has 2451 samples, and 20% has 613 samples. As shown in Figure 7, it illustrates the visualization of a percentage of the data used for training and testing purposes. There are many approaches used in measuring performance [60]. A confusion matrix is a tabular representation used to assess the performance of a classification model. It summarizes the number of correct and incorrect predictions made by the model, categorized by class [61]. 

The rows represent the actual classes, while the columns represent the predicted classes. The diagonal elements show the correct predictions for each class, while the nondiagonal elements show incorrect predictions. Several metrics, such as accuracy, precision, recall, and F1 score, can be calculated based on the confusion matrix. As shown in Figure 8, it shows the confusion matrix for a brain tumor MRI from the proposed solution. The correct classifications for each meningioma, glioma, and pituitary tumor are 132, 282, and 193, respectively. The number of glioma samples is 1426, compared with 708 and 930 for meningioma and pituitary tumors, respectively.

### Evaluation Metrics

To compare the results of the proposed model with those of the previous model, we need to analyze their performance on a typical set of benchmarks. By doing so, we can determine whether or not the proposed model outperforms the previous model. Performance measurements are crucial for evaluating the effectiveness of a classification model, allowing for comparisons between models to determine the best option. Common performance measurements include accuracy, precision, recall, F1-score, and area under the curve (AUC) [62]. Accuracy measures the proportion of correctly classified instances, as shown in Equation (7), while precision and recall measure the proportion of true positives among all positive predictions and actual positives, respectively. The F1 score balances precision and recall.
(9)$$Accuracy=\frac{TP+TN}{TP+FP+TN+FN}$$
where these symbols indicate true positive (*TP*), true negative (*TN*), false positive (*FP*), and false negative (*FN*).

Equation (2) shows how to calculate precision:(10)$$Precision=\frac{TP}{TP+FP}$$

Equation (3) presents the calculation of the recall value:(11)$$Recall=\frac{TP}{TP+FN}$$

The average between precision and recall is F1-measurement, which is calculated with Equation (4):(12)$$F1=\frac{2\times precision\times recall}{precision\times recall}$$

The following Table 5 and Figure 9 note our classification results for precision, recall, f1-score, and accuracy.

The AUC chat metric comprehensively assesses the DCTM model’s performance by considering the trade-off between true positive and false positive rates. It measures the model’s ability to correctly distinguish between positive and negative instances in the dataset. A higher AUC value indicates better discriminatory power, suggesting the model achieves higher positive rates while minimizing false positives. This evaluation metric is beneficial in scenarios where imbalanced classes exist, ensuring that a dominant class does not skew the model’s performance. Consequently, the AUC chat enables a reliable comparison of different models and aids in identifying the optimal threshold for classification decisions. The AUC of the proposed model’s receiver operating characteristic (ROC) was achieved at 0.99 as shown in Figure 10, which illustrates the AUC chat for the DCTM model.

## 6. Discussion

We compared our model with state-of-the-art models on the same dataset for consistent performance measurement. Numerous experiments have been conducted on the same dataset that we employed in our study; however, the models utilized varied, with some experiments using machine learning algorithms, such as SVM, Random Forest, Naïve Bayes, and KNN, while others relied on deep learning models based on their respective experimental requirements. Furthermore, there was a distinction in the type of classification employed, with some experiments utilizing binary classification (tumor or no tumor). In contrast, others employed multiclass classification (glioma, meningioma, pituitary tumor, and sometimes non-tumor is a plus) [63]. Notably, multiclass classification was the most commonly used method, consistent with our experimental approach. The following table compares our dual tumor convolution network results enhanced compared with other models in the literature as shown in Table 6.

By considering the range of models employed in previous studies, we aimed to comprehensively evaluate our model’s performance within the broader context of existing approaches. This approach allows for a better understanding of the strengths and limitations of our proposed method in comparison to both traditional machine learning algorithms and cutting-edge deep learning models used in the field.

## 7. Conclusions and Future Work

The Dual Convolution Tumor Network model is designed to achieve high accuracy in classification compared to previous models in the same research field. Initially, several architectures were considered, including EfficientNet, LetNet, Alex Net, and VIT, using different layers. Still, our proposed model, which consists of a pre-trained CNN model VGG16 and a custom convolutional network as the second layer, reached the highest accuracy of 99%. GlobalMaxPooling2D layers were added after each CNN layer before the fully connected layers, allowing for extracting important features and reducing the dimensionality of feature maps. The two CNNs were concatenated and followed by three fully connected layers. The first fully connected layer contained 1000 neurons and a dropout layer with a 0.3 dropping rate to address overfitting. The second fully connected layer contains 256 neurons, followed by another fully connected layer containing 64 neurons and a SoftMax output layer. The ReLU activation function is used in the network, and the loss function used is sparse and categorical. The experiments showed that the proposed dual convolution tumor network model achieved the highest accuracy compared to other state-of-the-art models.

Furthermore, in the future, we aim to propose a model for classifying glioma grades with an auto-segmentation feature used in image processing and computer vision, where an algorithm automatically segments or separates different regions of an image into distinct areas. Glioma grade classification is vital for guiding treatment decisions and predicting patient outcomes. Treatment options for gliomas include surgery, radiation therapy, chemotherapy, and targeted therapy, which vary depending on the tumor grade and other factors such as the patient’s age and overall health.

## Figures and Tables

**Figure 1 diagnostics-13-02050-f001:**
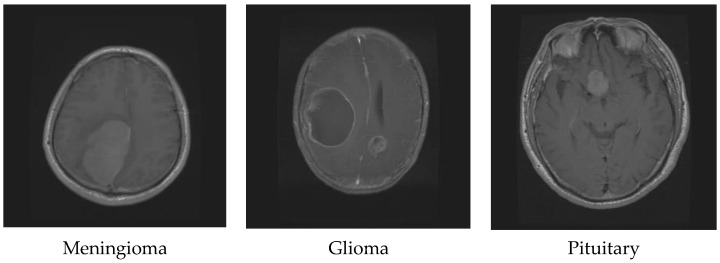
Different brain tumor types.

**Figure 2 diagnostics-13-02050-f002:**
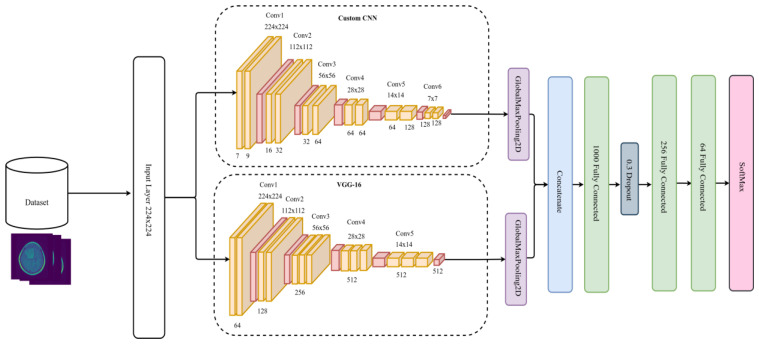
DCTN model architecture.

**Figure 3 diagnostics-13-02050-f003:**
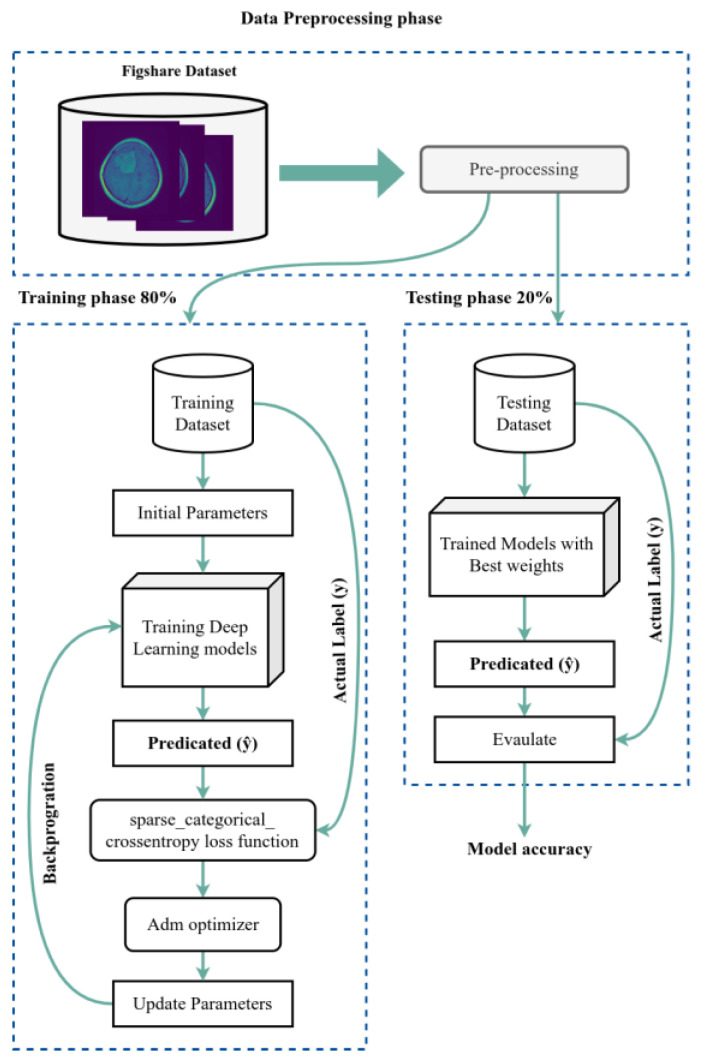
The training and testing process.

**Figure 4 diagnostics-13-02050-f004:**
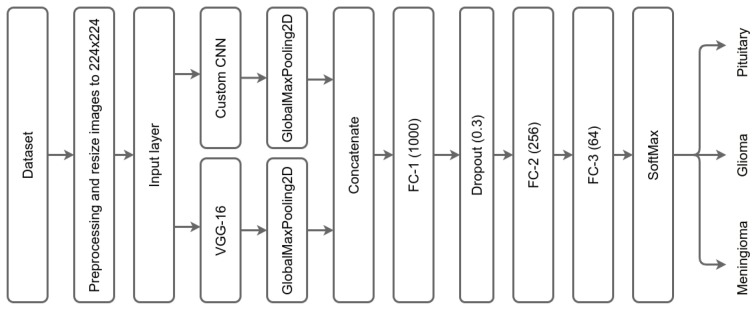
Summary of DCTN model layers.

**Figure 5 diagnostics-13-02050-f005:**
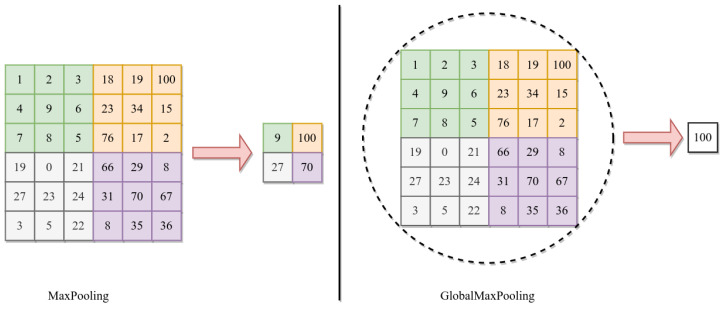
Difference between Maxpooling and Globalmaxpooling.

**Figure 6 diagnostics-13-02050-f006:**
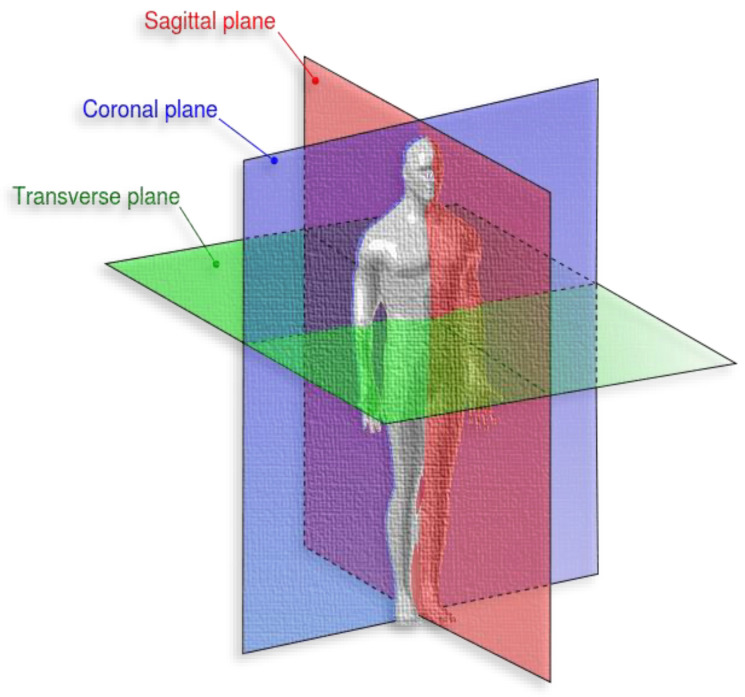
Different brain tumor types [57].

**Figure 7 diagnostics-13-02050-f007:**
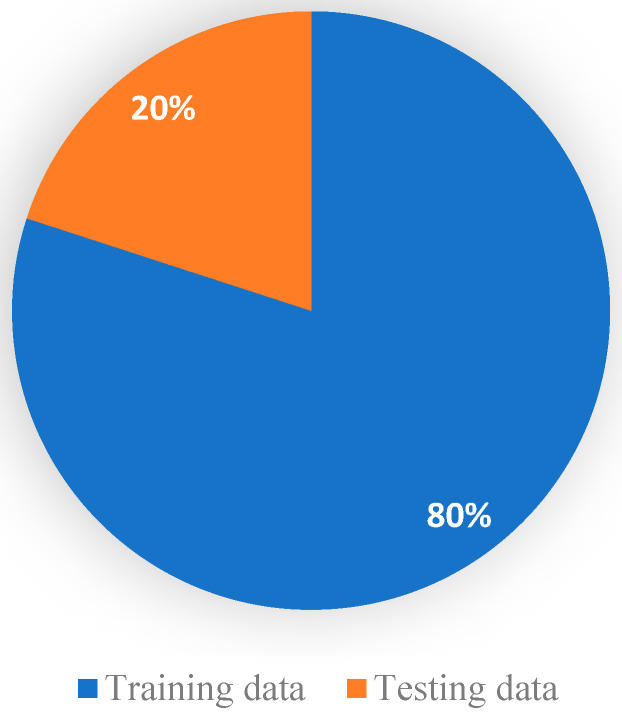
Visualization of a percentage of data used for training and testing.

**Figure 8 diagnostics-13-02050-f008:**
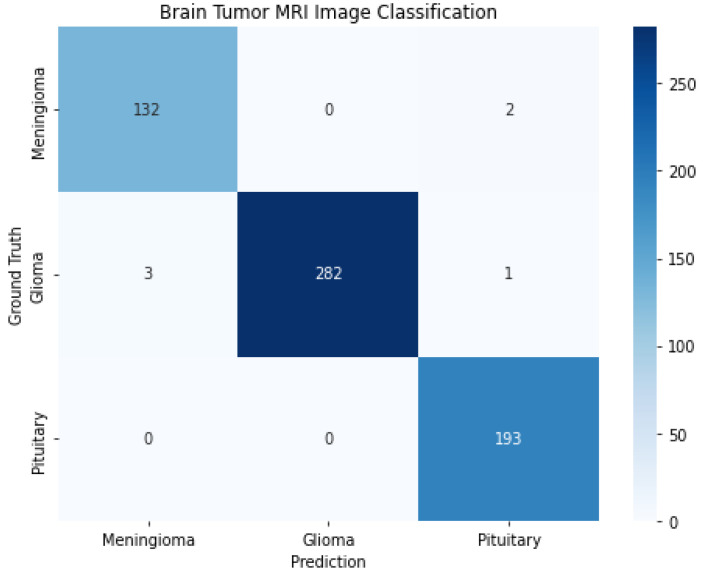
Confusion matrix for brain tumor MRI.

**Figure 9 diagnostics-13-02050-f009:**
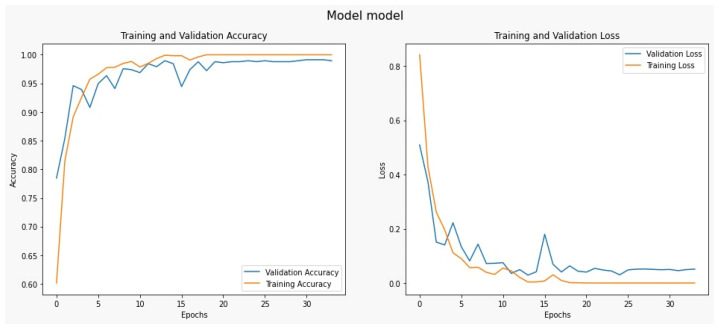
Classification performance for the proposed model.

**Figure 10 diagnostics-13-02050-f010:**
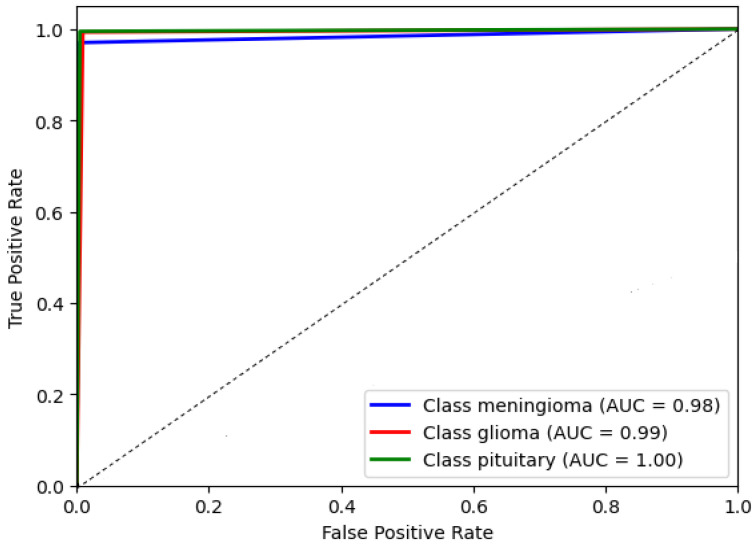
The proposed system evaluation.

**Table 1 diagnostics-13-02050-t001:** Summary of the VGG-16 layers.

	Layer	Input Size	No. Filters	Kernel Size	Stride
VGG-16	Conv1-1	224 × 224	64	3 × 3	2
	Conv1-2	224 × 224	64	3 × 3	
	Maxpool			2 × 2	2
	Conv2-1	112 × 112	128	3 × 3	2
	Conv2-2	112 × 112	128	3 × 3	2
	Maxpool			2 × 2	2
	Conv3-1	56 × 56	256	3 × 3	2
	Conv3-2	56 × 56	256	3 × 3	2
	Conv3-3	56 × 56	256	3 × 3	2
	Maxpool			2 × 2	2
	Conv4-1	28 × 28	512	3 × 3	2
	Conv4-2	28 × 28	512	3 × 3	2
	Conv4-3	28 × 28	512	3 × 3	2
	Maxpool			2 × 2	2
	Conv5-1	14 × 14	512	3 × 3	2
	Conv5-2	14 × 14	512	3 × 3	2
	Conv5-3	14 × 14	512	3 × 3	2
	Maxpool			2 × 2	2

**Table 2 diagnostics-13-02050-t002:** Summary of the custom network layers.

	Layer	Input Size	No. Filters	Kernel Size	Stride
**Custom CNN**	Conv1-1	224 × 224	7	3 × 3	1
	Conv1-2	224 × 224	9	3 × 3	1
	Maxpool			2 × 2	2
	Conv2-1	112 × 112	16	3 × 3	1
	Conv2-2	112 × 112	32	3 × 3	1
	Maxpool			2 × 2	2
	Conv3-1	56 × 56	32	3 × 3	1
	Conv3-2	56 × 56	64	3 × 3	1
	Maxpool			2 × 2	2
	Conv4-1	28 × 28	64	3 × 3	1
	Conv4-2	28 × 28	64	3 × 3	1
	Maxpool			2 × 2	2
	Conv5-1	14 × 14	64	3 × 3	1
	Conv5-2	14 × 14	128	3 × 3	1
	Conv5-1	7 × 7	128	3 × 3	1
	Conv5-2	7 × 7	128	3 × 3	1
	Maxpool			2 × 2	2

**Table 3 diagnostics-13-02050-t003:** Parameters used in training the DCTN model.

Parameters	Value
Initial learning rate	0.0001
Batch size	64
No. of epochs	50
Shuffle	every epoch
Loss function	Sparse-categorical-cross-entropy

**Table 4 diagnostics-13-02050-t004:** Dataset category description.

Tumor Type	Coronal	Axial	Sagittal	Total
Meningioma	268	209	231	708
Glioma	437	494	495	1426
Pituitary	319	291	320	930
**Total**				3064

**Table 5 diagnostics-13-02050-t005:** Classification report.

	Precision	Recall	F1-Score	Support
Meningioma	0.99	0.98	0.98	134
Glioma	0.99	1.00	0.99	286
Pituitary	0.98	0.99	0.99	193
Accuracy			0.99	613
Macro avg	0.99	0.99	0.99	613
Weighted avg	0.99	0.99	0.99	613

**Table 6 diagnostics-13-02050-t006:** Comparison with our results and related work’s results.

Method	Classification Type	Used Technique	Accuracy
In Ref. [63]	Multiclass (Glioma, Meningioma, and Pituitary Tumor)	CNN, Transfer learning (Google Net)	98%
In Ref. [64]	Multiclass (Glioma, Meningioma, and Pituitary Tumor)	CNN, Fine-tuned EfficientNetB2	98.86%
In Ref. [65]	Multiclass (Glioma, Meningioma, and Pituitary Tumor, Healthy Images)	Novel deep residual and regional-based Res-BRNet convolutional neural network (CNN)	98.22%
In Ref. [66]	Multiclass (Glioma, Meningioma, and Pituitary Tumor)	DenseNet201-based transfer learning MobileNet	98.22% 97.87%
In Ref. [67]	Binary-Classification (Malignant and Non-Malignant)	K-nearest neighbor (KNN) multiclass support vector machine (MSVM) neural network (NN)	88.43% 92.5% 95.86%
In Ref. [68]	Multiclass (Glioma, Meningioma, and Pituitary Tumor)	A siamese neural network called D-CNN, material recognition neural networks (MAC-CNN)	92.8%
In Ref. [69]	Multiclass (Glioma, Meningioma, Pituitary Tumor, and non-tumor)	Generative adversarial network (GAN), multiscale gradient GAN (MSGGAN) with auxiliary classification	98.57%
In Ref. [70]	Multi-Class (Glioma, Meningioma, and Pituitary Tumor)	CNN, cross-validation technique	96.56%
In Ref. [42]	Multiclass (Glioma, Meningioma, and Pituitary Tumor)	CNN includes a multiscale approach	97.3%
In Ref. [71]	Multiclass (Glioma, Meningioma, and Pituitary Tumor)	Brain tumor segmentation and classification network (BTSCNet)	Meningioma 96.6% (using MR-Contrast feature) Glioma 98.1% (using MR-Correlation feature) Pituitary 95.3% (using MR-Homogeneity feature)
Proposed Model	Multiclass (Glioma, Meningioma, and Pituitary Tumor)	Dual CNN (VGG16, Custom CNN)	99%

## Data Availability

All data generated or analyzed during this study are included in this published article.

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
