# Peer review of "Dual Deep CNN for Tumor Brain Classification"

_diagnostics, 2023, doi:10.3390/diagnostics13122050_

Round 1

Reviewer 1 Report

Manuscript Number: 2394450

Comments:

The article paper by Aya M. Al-Zoghby and colleagues entitled "Dual Deep CNN for Tumor Brain Classification" provides a great work of current knowledge and characterizations is complete, and the description is in good shape to prove the current data and is useful for the field. There are a couple of corrections that should be made and some suggestions to improve the clarity of the manuscript:

1.    In the introduction section, second paragraph, the author should select a newly published article or review papers as a reference, for example, adding references like L Pang et al. 2022 "Mechanism and therapeutic potential of tumor-immune symbiosis in glioblastoma" and "Pharmacological targeting of the tumor immune symbiosis in Glioblastoma." Please describe why it is essential to this study stands out; introducing dissecting phenotypic and genetic heterogeneity at spatial and temporal levels is highly challenging, and the dynamics of the GBM microenvironment cannot be captured by analysis of a single tumor sample, particularly in GBM.

2.    A recent reference elaborated by F Khan et al. 2023 "Macrophages and microglia in glioblastoma: heterogeneity, plasticity, and therapy."  on the GBM TME that contributes to most GBM hallmarks, including immunosuppression and the understanding of the heterogeneity of GBM has been limited by the lack of powerful tools to characterize. Adding explaination of how artificial intelligence can help to uncover brain tumor heterogeneity to target specifically, explain in the introduction section with the reference.  

3.    It is essential to consider, including briefly, the challenges to investigating artificial intelligence (AI) and how it plays a vital role in understanding disease and its various types of cancer. The authors could consider including this and perhaps briefly mentioning the challenges to investigating this.

4.    The overall article description is concise. I recommend that the author should elaborate briefly for the reader to understand the AI scope better in cancer and try to include recent publications as a reference.

5.    I would like to suggest that the authors have bioinformatics analysis, as they mentioned artificial intelligence. Having some analysis would strengthen the paper significantly.

Fine, but need to check for grammar.

Author Response

Dear Prof,

We would like to express our sincere appreciation for the constructive comments provided on our manuscript. We appreciate the constructive comments on the manuscript. Please find below the detailed answers to comments and see the highlighted file after taking into consideration all changes that have been required from all reviewers. We improved the structure as suggested to improve the quality of the paper. We did our best to make the paper in the optimal form for publication. We are grateful for the fruitful comments and the time and effort invested by the reviewers in evaluating our work. Their insights have been invaluable in enhancing the manuscript. We believe that the revised version effectively addresses the concerns raised and significantly improves the overall quality of the paper. Thank you for your time and consideration.

Best regards,

Authors

Reviewer 2 Report

Congratulations to the authors for their detailed analysis of a novel DL algorithm on CNS tumor radiomics.  In the enclosed PDF file you will find my extensive and particular comments.  My overall comments are the following:

1) There should be a slight improvement in the used language to meet the requirements of medical jargon

2) The Introduction section is rather extensive and thus should be made more concise.

3) There should be two separate and distinct Results and Discussion sections.

4) The AUROC metric should be calculated and added in the Results section.

The quality of language is generally good, but at some points, it seems that it does not address medical professionals and thus should be improved in this regard.

Author Response

(The authors gave the same response as above.)
